# Distributionally Robust Reinforcement Learning

Elena Smirnova [1]   Elvis Dohmatob [1]   Jeremie Mary [1]

## Abstract

Real-world applications require RL algorithms to act safely. During learning process, it is likely that the agent executes sub-optimal actions that may lead to unsafe/poor states of the system. Exploration is particularly brittle in high-dimensional state/action space due to increased number of low-performing actions. In this work, we consider risk-averse exploration in approximate RL setting. To ensure safety during learning, we propose the *distributionally robust policy iteration* scheme that provides tight lower bound guarantee on state-values. Our approach induces a dynamic level of risk to prevent poor decisions and yet preserves the convergence to the optimal policy. Our formulation results in a tractable algorithm that accounts for a simple re-weighting of policy actions in the standard policy iteration scheme. We extend our approach to continuous state/action space and present a practical algorithm, *distributionally robust soft actor-critic*, that implements a different exploration strategy: it acts conservatively at short-term and it explores optimistically in a long-run. We provide promising experimental results on continuous control tasks.

## 1. Introduction

At root of difficulties to deploy Reinforcement Learning in the real-world is the problem of exploration. In all generality, in order to guarantee the optimality of a policy we need to build estimates of all state-values that may result in some occasional catastrophic outcomes. This is of course unacceptable in many applications, such as real-world robot tasks (Abbeel & Ng, 2005) or online recommendation systems (Theocharous et al., 2015). One strategy to avoid disastrous events is to lower the risk in face of uncertainty.

In this work, we consider risk-averse exploration in the con-

[1]Criteo AI Lab. Correspondence to: Elena Smirnova <e.smirnova@criteo.com>.

*Reinforcement Learning for Real Life (RL4RealLife) Workshop in the $36^{th}$ International Conference on Machine Learning*, Long Beach, California, USA, 2019. Copyright 2019 by the author(s).

text of modified policy iteration (MPI) scheme (Puterman, 1994). MPI defines an iterative process that alternates between policy improvement and (partial) policy evaluation steps. Applying this scheme to practical problems with large state/action space and finite number of interactions leads to errors at the policy evaluation step, resulting in the approximate MPI scheme (Scherrer et al., 2015). In the approximate MPI the policy state-values are inexact, and thus, exploration strategies, such as Boltzmann exploration, are likely to execute poor actions (Garcıa & Fernández, 2015).

Prior works studied safety guarantees w.r.t. approximate policy evaluation step. Risk-sensitive approach explicitly modifies the policy's long-term outcome to incorporate the notion of risk, typically expressed as variance of return over policy trajectories. In approximate dynamic programming, risk-sensitive counterparts of value iteration and policy iteration have been developed (Tamar et al., 2013; Prashanth & Ghavamzadeh, 2016), although they are known to result in computationally difficult algorithms (Mannor & Tsitsiklis, 2011). In model-free setting, the proposed algorithms are computationally extensive as they involve integration over the state space and nonconvex parameter optimization. To mitigate this issue, approximation schemes have been proposed based on temporal differences (Mihatsch & Neuneier, 2002), stochastic approximation (Borkar, 2002) and recently, policy gradient (Tamar et al., 2015).

In this paper, we consider the inexact computation of policy state-values due to the finite number of interactions with the environment, called *estimation errors*. The risk-averse strategy consists in lowering the risk of catastrophic outcome under estimation errors. To implement risk-averse strategy, we introduce a family of *distributionally robust Bellman operators* that provide tight lower bound guarantee on policy state-values. Using this operator instead of standard Bellman operator, we formulate a *distributionally robust modified policy iteration* scheme that places an adaptive level of conservatism w.r.t estimation errors at policy evaluation step. Differently from prior work, our proposed algorithms are computationally tractable and preserve convergence to the optimal policy at growing amount of collected experience.

Using the Legendre-Fenchel transform (Boyd & Vandenberghe, 2004), our formulation results in a simple modification to the standard policy iteration scheme that consists in

computing the evaluation step w.r.t. re-weighted policy state-action probabilities. The proposed algorithm is applicable to large state spaces since the additional computation scales with the size of the action space. For continuous action space, we derive an efficient approximation of our scheme that only involves a constant-time reward modification.

We propose a practical algorithm for continuous control tasks, called *distributionally robust soft actor-critic*, by combining risk-averse policy evaluation under finite sample of data with optimistic exploration of Soft Actor-Critic (Haarnoja et al., 2018b). Distributionally robust soft actor-critic implements a different exploration strategy: it acts conservatively at short-term to ensure the lower bound on policy performance, and it explores optimistically at long-term to preserve convergence to the optimal policy.

To summarize, our main contributions are as follows:

- We propose a principled and scalable modification of MPI that ensures risk-averse policy evaluation w.r.t. estimation errors, while preserving convergence to the optimal policy. Convergence rate is analyzed.

- We apply this scheme to maximum entropy policies that results in a risk-averse short-term and optimistic long-term exploration strategy.

- We derive an extension of our scheme to the continuous state/action space that only involves a modification of reward. We provide promising experimental results on continuous control tasks.

## 2. Preliminaries

We will use the following notation: $\Delta_X$ is the set of probability distributions over a finite set $X$ and $Y^X$ is a set of mappings from set $X$ to set $Y$.

We consider a Markov decision process $M := (\mathcal{S}, \mathcal{A}, P, r, \gamma)$ where $\mathcal{S}$ is a finite state space, $\mathcal{A}$ is a finite action space, $P \in \Delta_{\mathcal{S}}^{\mathcal{S} \times \mathcal{A}}$ is the transition kernel with transition probability $p(s'|s,a)$, $r(s,a) \in \mathbb{R}^{\mathcal{S} \times \mathcal{A}}$, $|r(s,a)| \leq R_{\max}$ is a bounded reward function. We define a stochastic stationary policy $\pi \in \Delta_{\mathcal{A}}^{\mathcal{S}}$ and let $\Pi$ be the set of stochastic stationary policies. We consider the discounted setting with discount factor $\gamma \in [0, 1)$.

We define the Bellman operator $\mathcal{T}^\pi$ for any function $V \in \mathbb{R}^{\mathcal{S}}$, $\forall s \in \mathcal{S}$ as follows:

$$[\mathcal{T}^\pi V](s) := \mathbb{E}_{a \sim \pi(\cdot|s)} \left[ r(s,a) + \gamma \mathbb{E}_{s' \sim p(s'|s,a)}[V(s')] \right] \tag{1}$$

This is a $\gamma$-contraction in $\ell_\infty$ norm and its unique fixed point is $V^\pi$: $\lim_{k \to \infty} (\mathcal{T}^\pi)^k V = V^\pi$, where equality holds component-wise. By denoting $Q_V(s,a) := r(s,a) +$

$\gamma \mathbb{E}_{s' \sim p(s'|s,a)}[V(s')]$, Eq. (1) can be re-written as:

$$[\mathcal{T}^\pi V](s) = \langle \pi(\cdot|s), Q_V(s, \cdot) \rangle. \tag{2}$$

From (1), we define the Bellman optimality operator as follows:

$$[\mathcal{T}^\star V](s) := \max_{\pi(\cdot|s) \in \Delta(\mathcal{A})} [\mathcal{T}^\pi V](s) \ \forall s \in \mathcal{S}, \tag{3}$$

which is a $\gamma$-contraction in $\ell_\infty$ norm and its unique fixed point is the optimal value function $V^\star$. Further, the equalities hold state-wise, so we will omit the per-state notation and write $\mathcal{T}^\star V = \max_{\pi \in \Pi} \mathcal{T}^\pi V$.

We denote by $\mathcal{G}(V)$ the set of optimal policies that achieve the maximum of Eq. (3) state-wise:

$$\mathcal{G}(V) := \{ \pi : \pi \in \arg\max_{\pi \in \Pi} T^\pi V \} \tag{4}$$

Equivalently, this set coincides with the set of optimal policies: $\mathcal{G}(V) = \{ \pi : \mathcal{T}^\pi V = \mathcal{T}^\star V \}$.

The Modified Policy Iteration (MPI) algorithm (Puterman, 1994) is the iterative process that alternates between policy improvement and (partial) policy evaluation steps:

$$\pi_{t+1} \in \mathcal{G}(V_t); \ V_{t+1} = (\mathcal{T}^{\pi_{t+1}})^m V_t \tag{5}$$

where $m = 1$ corresponds to Value Iteration and $m = \infty$ corresponds to Policy Iteration. Here $V_t$ denotes an approximation of $V^{\pi_t}$.

Finally, we will make use of the Legendre-Fenchel duality, e.g. see Section 3.3.1 in (Boyd & Vandenberghe, 2004). For a strongly convex function $\Omega : \Delta(\mathcal{A}) \to \mathbb{R}$ its Fenchel dual $\Omega^\star : \mathbb{R}^{\mathcal{A}} \to \mathbb{R}$ is given by:

$$\Omega^\star(Q_V) = \max_{\pi \in \Delta(\mathcal{A})} \langle \pi, Q_V \rangle - \Omega(\pi). \tag{6}$$

Using properties of the Fenchel transform at the maximum of the of (6) we have for the gradient of the dual function:

$$\nabla \Omega^\star(Q_V) = \arg\max_{\pi \in \Delta(\mathcal{A})} \langle \pi, Q_V \rangle - \Omega(\pi) \tag{7}$$

Similarly to standard Bellman operators, we define the regularized Bellman operator (Geist et al., 2019) as follows:

$$\mathcal{T}^{\pi, \Omega} V := \mathcal{T}^\pi V - \Omega(\pi). \tag{8}$$

and the set of optimal policies:

$$\mathcal{G}^\Omega(V) := \{ \pi : \pi \in \arg\max_{\pi \in \Pi} \mathcal{T}^{\pi, \Omega} V = \nabla \Omega^\star(Q_V) \} \tag{9}$$

# 3. Distributionally robust policy iteration

We consider the Approximate Modified Policy Iteration scheme (Scherrer et al., 2015), where the evaluation step in (5) is subject to an estimation error $\delta_t$ due to finite sample of transitions used to perform evaluation:

$$\pi_{t+1} \in \mathcal{G}(\tilde{V}_t); \ \tilde{V}_{t+1} = \mathcal{T}^{\pi_{t+1}} \tilde{V}_t + \delta_t. \qquad (10)$$

This is indeed a practical scenario as state-of-the-art off-policy algorithms sample a mini-batch of independent samples from a replay buffer to perform value update (Mnih et al., 2013); on-policy algorithms directly draw a finite number of trajectories from $\pi_{t+1}$ (Schulman et al., 2015).

Due to the finite amount of collected experience, the value $V_t$ is uncertain. The uncertainty of empirical estimate is captured by its variance and is known under *parametric uncertainty* (Mannor et al., 2004). Optimistic exploration strategies (Auer et al., 2002; Jaksch et al., 2010) are classically used under parametric uncertainty. They construct, at any state, an optimistically biased value estimate and take action with the highest value. If the selected action is not near-optimal, i.e., the value estimates are overly-optimistic, it may lead to unsafe states of the system. Therefore, to prevent the exploration process from catastrophic outcomes, we consider the notion of safety under finite sample estimates.

In Section 3.1 we introduce a family of Bellman operators that guarantee policy performance with tight finite-sample bounds. Section 3.2 describes their efficient computation and the resulting policy iteration scheme. Theorem 1 establishes the asymptotic convergence of the proposed scheme to the optimal policy, while reducing the chance of catastrophic failure. In Section 3.3 we apply this scheme to a class of maximum entropy policies. The resulting algorithm is expected to achieve, under parametric uncertainty, a risk-averse behaviour at short-term time horizon and a risk-seeking behaviour in a long-run.

## 3.1. Distributionally robust Bellman operator

We formalize the risk-averse strategy under estimation errors in approximate MPI scheme (10). We define a family of operators that represent a lower bound on the exact operator given a sequence of convergent errors.

**Definition 1** (Distributionally robust Bellman operator). *For a sequence of error vectors $\epsilon_1, \epsilon_2, \ldots, \epsilon_t, \ldots \in \mathbb{R}^{\mathcal{S}}$, we say that an operator $\mathcal{T}^{\pi_t}_{\epsilon_t} : \mathbb{R}^{\mathcal{S}} \to \mathbb{R}^{\mathcal{S}}$ is distributionally robust if*

- $\mathcal{T}^{\pi_t}_{\epsilon_t}$ *is a Bellman operator and provides a tight lower bound on $\mathcal{T}^{\pi_t}$ at finite time:*

$$\mathcal{T}^{\pi_t}_{\epsilon_t} v \leq \mathcal{T}^{\pi_t} v \leq \mathcal{T}^{\pi_t}_{\epsilon_t} v + \epsilon_t, \ \forall v \in \mathbb{R}^{\mathcal{S}}. \qquad (11)$$

- *The errors $\epsilon_t$ are convergent:*

$$\limsup_{N \to \infty} \sum_{t=1}^{N-1} \gamma^t \|\epsilon_{N-t}\|_{\infty} = 0. \qquad (12)$$

This Bellman operator provides robustness w.r.t finite sample of observations, represented by the estimation error $\epsilon_t$ that should decrease with the amount of collected experience. Condition (12) on the sequence of errors $\epsilon_t$ implies an asymptotic convergence to the exact evaluation step of (5). The next lemma formally introduces this statement.

**Lemma 1** ( (Scherrer et al., 2015)). *For any initial value function $V_0$ consider the approximate MPI $\tilde{V}_t = (\mathcal{T}^{\pi_t})^m \tilde{V}_{t-1} + \delta_t$ (with $\tilde{V}_0 = V_0$); $\pi^{t+1} \in \mathcal{G}(V_t)$. Then, one has*

$$\|\tilde{V}_N - V^*\|_{\infty} \leq \frac{4 R_{\max}}{(1-\gamma)^2} E_N + \frac{2\gamma^N}{1-\gamma} \|V_0 - V^*\|_{\infty},$$

*where $E_N := \sum_{t=1}^{N-1} \gamma^t \|\delta_{N-t}\|_{\infty}$.*

**Remark.** *By virtue of Def. 1, the approximate MPI using distributionally robust operator is convergent if the sum of discounted uncertainties satisfies $\limsup_{N \to \infty} \sum_{t=1}^{N-1} \gamma^t \|\epsilon_{N-t}\|_{\infty} = 0$. This is because $|\delta_t(s)| \leq \square \epsilon_t(s) \ \forall s$ for some global constant $\square$, and so $\limsup_N E_N \leq \square \limsup_N \sum_{t=1}^{N-1} \gamma^t \|\epsilon_{N-t}\|_{\infty} = 0$.*

Later, we will consider specific constructions of the error sequences in the form $\epsilon_t \propto n_t(s)^{-\eta}$, for some $\eta > 0$, where $n_t(s)$ is a state visitation count. We will show that for this construction of error sequence, the approximate MPI scheme with distributionally robust Bellman operator converges to the optimal policy-value pair (see Theorem 1).

One way to implement the distributionally robust Bellman operator is to consider the worst-case outcome of executing each action. Indeed, placing adequate uncertainty around action probabilities prevents the agent from selecting possibly low-value actions. We implement this idea in a variant of distributionally robust Bellman operator.

**Definition 2** (Uncertainty set and adversarial Bellman operator). *Given a policy $\pi$ and an error function $\epsilon \in \mathbb{R}^{\mathcal{S}}$, define the uncertainty set $\mathcal{U}_\epsilon(\pi)$ by*

$$\mathcal{U}_\epsilon(\pi) := \{\tilde{\pi} \in \Delta_{\mathcal{A}}^{\mathcal{S}} \mid D_{\mathrm{KL}}(\tilde{\pi}(\cdot|s) \| \pi(\cdot|s)) \leq \epsilon(s), \ \forall s \in \mathcal{S}\}, \qquad (13)$$

*Also define the (KL-based) adversarial Bellman operator $\mathcal{T}^{\pi^\epsilon} : \mathbb{R}^{\mathcal{S}} \to \mathbb{R}^{\mathcal{S}}$ by*

$$\mathcal{T}^{\pi^\epsilon} V := \min_{\tilde{\pi} \in \mathcal{U}_\epsilon(\pi)} \mathcal{T}^{\tilde{\pi}} V. \qquad (14)$$

**Proposition 1.** $\mathcal{T}^{\pi^\epsilon}$ *is a valid Bellman operator (i.e a monotone $\ell_\infty$-norm $\gamma$-contraction mapping on value functions) $\mathcal{T}^{\pi^\epsilon}$. Moreover, $\mathcal{T}^{\pi^\epsilon}$ is distributionally robust.*

*Proof.* Since mapping $\mathcal{T}^{\pi^\epsilon}$ is continuous in $\pi$ and $\mathcal{U}_\epsilon$ is compact, it follows from the Extreme Value Theorem that there exists a policy $\pi^\epsilon \in \mathcal{U}_\epsilon$ such that $\mathcal{T}^\pi_\epsilon = \mathcal{T}^{\pi^\epsilon}$. Thus, $\mathcal{T}^{\pi^\epsilon}$ is a valid Bellman operator. In addition, from (14) there exists a policy $\pi^\epsilon \in \mathcal{U}_\epsilon$ such that $\mathcal{T}^{\pi^\epsilon} \leq \mathcal{T}^{\tilde{\pi}} \; \forall \tilde{\pi} \in \mathcal{U}_\epsilon(\pi_t)$ including $\pi_t$. This proves the left inequality in (11). The right inequality follows from Lemma 2 (see Appendix) using Pinsker's inequality. $\qquad\square$

The minimization problem of adversarial Bellman operator defines an adversarial policy $\pi^\epsilon$ to $\pi$ as the worst-case realization from the uncertainty set defined in terms of Kullback-Leibler (KL) divergence. The degree of adversarial behaviour is controlled by the size of uncertainty set.

### 3.2. Distributionally robust policy evaluation

In the following, we derive an efficient computation scheme of adversarial Bellman operator (14). We show that it results in simple analytical expression for adversarial policy where the robustification appears as re-weighting of samples based on the adversarial distribution.

First, we note that the KL constraint in (13) is separable by state and is strongly convex w.r.t. $\tilde{\pi}$. This makes possible to explicitly express the minimizer $\pi^\epsilon$ of (14), i.e. the worst-case policy from the uncertainty set, thanks to the Legendre-Fenchel transform (6). We proceed by applying strong Lagrangian duality and re-writing as maximization problem:

$$
\begin{aligned}
[\mathcal{T}^{\pi^\epsilon} V](s) = &\max_{\lambda(s)>0} \min_{\tilde{\pi} \in \Delta(\mathcal{A})} [\mathcal{T}^{\tilde{\pi}} V](s) \\
&+ \lambda(s) D_{\mathrm{KL}}(\tilde{\pi}(\cdot|s) \| \pi(\cdot|s)) - \lambda(s)\epsilon(s) \\
= &\min_{\lambda(s)>0} \max_{\pi \in \Delta(\mathcal{A})} [-\mathcal{T}^\pi V](s) \\
&- \lambda(s) D_{\mathrm{KL}}(\tilde{\pi}(\cdot|s) \| \pi(\cdot|s)) + \lambda(s)\epsilon(s)
\end{aligned}
\tag{15}
$$

where $\lambda(s)$ is a per-state positive Lagrange multiplier.

Next, using Fenchel duality (6) in the case of KL-divergence, the solution to the *inner* maximization problem of (15) is given by the gradient of Fenchel convex conjugate (7) :

$$
\pi^\epsilon(a|s; \lambda) \propto \exp(-Q_V(s,a)/\lambda(s))\pi(a|s) \tag{16}
$$

The analytical expression for the adversarial policy has the following interpretation: the adversarial policy re-weights the sampling policy such that the worst-case actions are taken more frequently to the extent determined by the adversarial temperature $\lambda(s)$. Infinitely small uncertainty set $\epsilon \to 0^+$ results in $\lambda \to +\infty$, i.e. the adversarial policy taking the same actions as the optimizing policy. On the other hand, a too large uncertainty set $\epsilon \to +\infty$ leads to conservative policies as $\lambda \to 0^+$.

To connect the radius of uncertainty set $\epsilon$ to the optimal state-dependent parameter $\lambda^\star(s)$, we consider the solution to the

*outer* Lagrangian dual problem in (15). By representing the inner maximization problem of (15) in terms of its Fenchel dual (6), we obtain:

$$
\lambda^\star(s) := \arg\min_{\lambda(s)>0} \Omega^\star(-Q_V/\lambda(s)) + \lambda(s)\epsilon(s). \tag{17}
$$

This formulation is a 1-d convex optimization over a per-state expression that defines the optimal level of adversarial behaviour.

We summarize the presented results in the following.

**Corollary 1.** *The adversarial Bellman operator (14) can be expressed as a regularized Bellman operator (8) w.r.t. adversarial policy (16) with optimal $\lambda^\star(s)$ defined in (17). The associated distributionally robust modified policy iteration scheme is given by:*

$$
\begin{cases}
\pi_{t+1} \leftarrow \mathcal{G}(\tilde{V}_t) \\
\tilde{V}_{t+1} \leftarrow (\mathcal{T}^{\pi_{t+1}^{\epsilon_t}})^m \tilde{V}_t
\end{cases}
\tag{18}
$$

We now analyze the convergence of this scheme. Theorem 1 states than any convergent and consistent Bellman operator iteration can be made distributionally robust using (18) and yet converges to the optimal policy in terms of the $\ell_\infty$ norm. The convergence rate of (18) is polynomial instead of exponential in exact MPI (5).

**Theorem 1** (Distributionally robust modified policy iteration)**.** *For an integer $m \in [1, \infty]$, consider the distributionally robust MPI scheme (18), where*

$$
\epsilon_t(s) = \begin{cases}
C n_t(s)^{-\eta}, & \text{if } n_t(s) \geq t/S, \\
0, & \text{else,}
\end{cases}
$$

*for some constants $C, \eta > 0$ and $S$ denotes the number of states.*

*Let $\tilde{\ell}_t := \tilde{V}_t - V^* \in \mathbb{R}^S$ be the loss at iteration $t$. Then after $N \geq 2$ iterations, we have*

*(A) **Sub-optimality bound:***

$$
\|\tilde{\ell}_N\|_\infty \leq \frac{4R_{\max}}{(1-\gamma)^2} E_N + \frac{2\gamma^N}{1-\gamma}\|\tilde{\ell}_0\|_\infty,
$$

*where*

$$
E_N := \sum_{t=1}^{N-1} \gamma^t \|\delta_{N-t}\|_\infty = \mathcal{O}_N\left(\frac{CS^\eta}{(1-\gamma)N^\eta}\right) \longrightarrow 0.
$$

*(B) **Safety guarantee:***

$$
\tilde{V}_t \leq V_t \leq V^*, \; \forall t \in \{1, \ldots, N\},
$$

*where $V_t$ is the value function computed via exact evaluation step (5).*

*Proof.* See Appendix A.1. □

**Remark.** *Parameters $C$ and $\eta$ define the size of uncertainty set, thus, the degree of robustness of the policy. The lower levels imply higher robustness and slower convergence.*

**Remark.** *The condition $n_t(s) \geq t/S$ ensures that states with too few visits are not considered for the uncertainty set construction. The choice $\eta = 1/2$ leads to the bound $E_N = \mathcal{O}_N\left(\frac{\sqrt{S}}{(1-\gamma)\sqrt{N}}\right)$. It is can be motivated by the general fact that empirical means (obtained from finite samples) are within $\mathcal{O}(1/\sqrt{N})$ of their expected values.*

The pseudo-code corresponding to scheme (18) is presented in Algorithm 1. To implement this algorithm, it is sufficient to learn on samples from adversarial policy. One possibility to do so is to re-sample transitions based on the target distribution, e.g., using importance sampling. Alternatively, one can directly generate samples from the adversarial policy.

### 3.3. Extension to entropy-regularized policies

We apply the distributionally robust approach to the maximum entropy framework (Ziebart et al., 2008; Haarnoja et al., 2017) that has been recently successful on a variety of simulated and real-world tasks (Haarnoja et al., 2018b). The maximum entropy objective modifies the standard RL objective by adding a per-state entropy bonus; it results in improved exploration targeted at high-value actions (Haarnoja et al., 2018a).

Differently, distributionally robust policy iteration scheme (18) ensures a robust behaviour in face of uncertainty. We combine the best of both worlds by applying the proposed scheme (18) to the class of entropy-regularized policies (Haarnoja et al., 2017).

We define *soft adversarial Bellman operator* as follows:

$$\mathcal{T}^{\pi^\epsilon, \Omega} V := \min_{\tilde{\pi} \in \mathcal{U}_\epsilon(\pi)} \mathcal{T}^{\tilde{\pi}, \Omega} V, \qquad (19)$$

where

$$\Omega(\pi(\cdot|s)) = \alpha \mathcal{H}(\pi(\cdot|s)) \; \forall s \in \mathcal{S}.$$

**Proposition 2.** *$\mathcal{T}^{\pi^\epsilon, \Omega}$ is a valid Bellman operator (i.e a monotone $\ell_\infty$-norm $\gamma$-contraction mapping on value functions). Moreover, $\mathcal{T}^{\pi^\epsilon, \Omega}$ is distributionally robust.*

*Proof.* $\mathcal{T}^{\pi, \Omega}$ is a $\gamma$-contraction w.r.t to the $\ell_\infty$-norm on value functions (Geist et al., 2019). Analoguously to $\mathcal{T}^{\pi^\epsilon}$, $\mathcal{T}^{\pi^\epsilon, \Omega}$ has a solution in the set of policies since $\mathcal{U}_\epsilon(\pi)$ is compact and $\mathcal{T}^{\pi, \Omega}$ is continuous in $\pi$. □

The solution to regularized maximization problem (9) at the policy improvement step is given by the gradient of Fenchel

dual (9), also referred in the literature to the Boltzmann policy:

$$\pi(a|s) \propto \exp(Q_V(s, a)/\alpha) \qquad (20)$$

where $\alpha > 0$ is the exploration temperature. Thus, the adversarial policy (16) takes the following form:

$$\pi^\epsilon(a|s; \alpha, \lambda) \propto \exp((1/\alpha - 1/\lambda(s))Q_V(s, a)) \qquad (21)$$

The resulting *soft distributionally robust modified policy iteration scheme* is given by:

$$\begin{cases} \pi_{t+1} \leftarrow \mathcal{G}^\Omega(\tilde{V}_t) \\ \tilde{V}_{t+1} \leftarrow (\mathcal{T}^{\pi_{t+1}^{\epsilon_t}, \Omega})^m \tilde{V}_t. \end{cases} \qquad (22)$$

This is the counterpart of scheme (18) for regularized policies. The difference lies in the presence of the regularizer $\Omega$ in both greedy and evaluation steps. Note that, as in (18), the evaluation step is performed w.r.t adversarial policy (21), while the greedy step is done w.r.t. Boltzmann policy (20).

The scheme (22) different in nature from Soft Q-learning (Haarnoja et al., 2017) and other entropy-based approaches (Nachum et al., 2017; Haarnoja et al., 2018a). Despite apparent similarity in automatic temperature tuning, the scheme (22) adjusts temperature to provide a lower-bound guarantee on policy state-values, while the above-mentioned entropy-based approaches adjust temperature to ensure a target entropy level alike a re-parametrization. Moreover, the temperature adjustment in our scheme (22) is only performed at the policy evaluation step.

Since approximate regularized MPI shares the same error propagation bounds as unregularized MPI according to Corollary 1 of (Geist et al., 2019), the convergence of scheme (22) is an adaptation of Theorem 1.

---

**Algorithm 1** Distributionally Robust Policy Iteration

Initialize $V$, counters $n = 0$            ▷ Initialize
Set $C, \eta > 0$
**repeat**
    $\pi \leftarrow \mathcal{G}(V)$             ▷ Maximizing policy
    $\epsilon \leftarrow Cn^{-\eta}$         ▷ Size of uncertainty size
    $\lambda \leftarrow$ optimization of (17) ▷ Regularization parameter
    $\pi^\epsilon \propto \exp(-Q_V/\lambda)\pi$     ▷ Adversarial policy
    $V \leftarrow \mathcal{T}^{\pi^\epsilon} V$     ▷ Adversarial Bellman operator
**until** convergence

---

## 4. Continuous control

Real-world applications frequently operate in continuous state/action space. To extend the distributionally robust approach to continuous setup, we need to efficiently compute the adversarial Bellman operator in (18). Thus, in

Section 4.1 we derive an approximated scheme for KL-regularized Bellman operators that only involves a modification of reward. This allows us to formulate *distributionally robust soft actor-critic*, a practical algorithm that applies the soft distributional robustness scheme (22) to the soft actor-critic algorithm (Haarnoja et al., 2018b) (see Section 4.2).

### 4.1. KL-regularized Bellman operator

The regularized Bellman operator can be written in terms of its Fenchel conjugate (6): $[\mathcal{T}^\Omega V](s) = \Omega^\star(Q_V(s, \cdot))$. Define the regularization parameter $\lambda$ such that $\Omega_\lambda(\pi(\cdot|s)) := \lambda\Omega(\pi(\cdot|s))$ and consider the case of KL-divergence based regularization with respect to the prior policy $\mu \in \Delta_{\mathcal{S}}^{\mathcal{A}}$: $\Omega(\pi(\cdot|s)) = -D_{\mathrm{KL}}(\pi(\cdot|s)||\mu(\cdot|s))$. Then, the Fenchel conjugate is given by the smoothed maximum (minimum):

$$\Omega_\lambda^\star(Q_V(s, \cdot)) = \lambda \log \mathbb{E}_{a\sim\mu(\cdot|s)} \exp(Q_V(s, a)/\lambda) \quad (23)$$

for $\lambda > 0$ ($\lambda < 0$) respectively.

When the action space is continuous, the computation of the dual function (23) is intractable. To overcome the problem, we derive computationally feasible approximation of the smoothed maximum (minimum) function. We notice that smoothed maximum (minimum) can be seen as the logarithm of *moment-generating function* of the dual variable. By performing Taylor expansion of the moment-generating function around $\lambda \to +\infty$ ($\lambda \to -\infty$) and keeping terms up to the 1st order, we obtain:

$$\Omega_\lambda^\star(Q_V(s, \cdot)) = \mathbb{E}_{a\sim\mu}[Q_V(s, a)]$$
$$+ \frac{1}{2\lambda}\mathrm{Var}_{a\sim\mu}(Q_V(s, a)) + \mathcal{O}\left(\frac{1}{\lambda^2}\right). \tag{24}$$

This approximation gives a new perspective on KL-divergence regularized Bellman operators as encouraging ($\lambda > 0$) or penalizing variance ($\lambda < 0$) of Q-values under action distribution induced by the prior policy. The parameter $\lambda$ controls the amount of the regularization. In this view, distributionally robust Bellman operator can be seen as data-driven per-state variance penalization that adapts to the degree of uncertainty over the finite sample of data.

Approximation (24) provides an efficient way to compute the regularized Bellman operator through a simple modification of reward. Indeed, define potential function $\Phi(s) \in \mathbb{R}^{\mathcal{S}}$ as weighted variance of Q-values under the prior policy

$$\Phi(s) := \frac{1}{2\lambda}\mathrm{Var}_{a\sim\mu}(Q_V(s, a))$$

and a reward shaping function $r^\Omega(s, a, s') \in \mathbb{R}^{\mathcal{S}\times\mathcal{A}\times\mathcal{S}}$ as

$$r^\Omega(s, a, s') := r(s, a) + \gamma\Phi(s') - \Phi(s). \tag{25}$$

Then, by applying Corollary 2 of (Ng et al., 1999) Eq. (24) can be expressed using potential-based reward shaping:

$$\Omega_\lambda^\star(Q_V(s, \cdot)) \simeq Q_V(s, a) + \frac{1}{2\lambda}\mathrm{Var}_{a\sim\mu}(Q_V(s, a))$$
$$= \mathbb{E}_{a\sim\mu,s'\sim p(\cdot|s,a)}[r^\Omega(s, a, s') + \gamma V(s')]. \tag{26}$$

For $\lambda < 0$, the reward shaping function (25) encourages the policy to visit states with less variance over Q-values, i.e. expected to be "safer" states.

### 4.2. Distributionally robust soft actor-critic

We consider the class of entropy-regularized policies over continuous state/action space (Haarnoja et al., 2018a). We apply the soft distributionally robust policy iteration scheme (22) using reward modification (26) to approximate the computation of regularized Bellman operator over continuous action space.

As in (Haarnoja et al., 2018a), we consider parametrized Gaussian policies $\pi_\theta(a|s) = \mathcal{N}(\mu_\theta(s), \Sigma_\theta^2(s))$. To compute a modified reward (26), we approximate the variance of Q-values in (26) using the 1st order Taylor approximation of Q-values around the mean action:

$$\mathrm{Var}_{a\sim\pi_\theta}(Q(s, a)) \simeq g_0(s)^T \Sigma_\theta(s) g_0(s) \tag{27}$$

where $g_0(s) = \nabla_a Q(s, a)|_{a=\mu_\theta(s)}$. This approximation involves computing the gradient of Q-values w.r.t. action evaluated at the mean action. When actions are independent $\Sigma_\theta(s) = diag(\sigma_{1,\theta}(s), \ldots, \sigma_{K,\theta}(s))$, the expression (27) simplifies to computing the norm of the gradient weighted by the variance of corresponding action dimension:

$$\mathrm{Var}_{a\sim\pi_\theta}(Q(s, a)) \simeq ||g_0(s)||^2_{diag(\sigma_{1,\theta}(s),\ldots,\sigma_{K,\theta}(s))} \tag{28}$$

We summarize the above steps in Algorithm 2.

## 5. Related work

**Robust MDP** In model-based setting, robust MDP framework has been proposed (Nilim & El Ghaoui, 2004; Iyengar, 2005) that optimizes over the worst-case realization of uncertain MDP parameters. Specifically, the dynamic programming approach is used to optimize a minimax criterion over an uncertainty set that contains possible MDPs defined in terms of their transition matrices. Multiple works report the worst-case criterion may lead to overly conservative policies (Mihatsch & Neuneier, 2002; Gaskett, 2003). Variants of robust MDP formulations have been proposed to mitigate the conservativeness when additional information on parameter distribution is present (Mannor et al., 2012; Xu & Mannor, 2010; Tirinzoni et al., 2018). Differently, in this work, we employ adaptive uncertainty sets that reflect

**Algorithm 2** Distributionally Robust Soft Actor-Critic

Initialize: actor $\pi_\theta$, critic $Q$
Set entropy level $\mathcal{H}$
Set $C, \eta > 0$
Initialize $s$
**for** number of epochs **do**
    **for** number of samples **do**
        $a \sim \pi_\theta(s), s' \sim p(s'|s, a)$
        $\mathcal{D} \leftarrow \mathcal{D} \cup \{(s, a, r, s')\}$
    **end for**
    **for** number of updates **do**
        $(s, a, r, s') \sim D$
        $\epsilon \leftarrow C n^{-\eta}(s)$
        $\lambda \leftarrow$ 1-d optimization of Eq. (17)
        $\sigma_Q^2(s) \leftarrow \sum_{i=1}^K g_0(s)^2 * \sigma_{i,\theta}^2$, see Eq. (28)
        $r^\Omega(s, a) \leftarrow r(s, a) + \frac{1}{2\lambda}(\gamma \sigma_Q^2(s') - \sigma_Q^2(s))$, see Eq. (25)
        $Q(s, a) \leftarrow r^\Omega(s, a) + \gamma(Q(s', \pi_\theta(s')) - \alpha \log \pi_\theta(s'))$, see Eq. (26)
        Update $\pi_\theta, \alpha$ as in Alg. 1 of (Haarnoja et al., 2018a)
    **end for**
**end for**

the amount of uncertainty associated with the finite sample size. We define our uncertainty set in terms of policies that arise naturally in stochastic iterative schemes and result in computationally efficient algorithms.

**Risk-sensitive MDP** The framework of risk-sensitive MDP optimizes a modified objective expressed using a risk-sensitive criterion, such as the expected exponential utility or variance-related measure, w.r.t. to the long-term policy performance. In the model-free context, risk-sensitive RL for expected exponential utility has been proposed in (Borkar, 2002) and for coherent risk measures (Tamar et al., 2015). The proposed algorithms are computationally extensive as they involve integration over state space and nonconvex parameter optimization. Differently, we consider a short-term dynamic risk that shrinks with the amount of collected data, and thus, preserves the desired level of long-term risk.

**Adversarial RL** Adversarial robustness in RL has been a focus of recent works. (Pinto et al., 2017) studied robustness to model parameters perturbations by manually engineering adversarial forces for a set of continuous control tasks. (Tessler et al., 2019) integrates the adversarial policy into the agent's policy definition through a convex combination or a noisy action perturbation. The proposed methods are formulated as instances of two-player zero-sum Markov game (Littman, 1994; Perolat et al., 2015). The distributionally robust approach is related to a regularized zero-sum Markov game (Geist et al., 2019), where the adversary is

|  | Hopper | Walker2D |
|---|---|---|
| Return Avg | -1.7%±89 | -0.4%±48 |
| Return Std | **-76%±21** | **-78%±48** |
| Episode Len Avg | +8%±82 | +4.8%±89 |
| Episode Len Std | **-76%±13** | **-77%±42** |

*Table 1.* Percent change of DR-SAC vs. SAC metrics computed over 100 test trajectories of the final policy. DR-SAC policies generate trajectories with smaller variance of return and episode length and similar average values.

regularized against the optimizing policy with a dynamic data-driven factor. Differently, it defines a multi-stage game where the first $m$ steps are played by the adversary.

## 6. Experiments

We compare the robustness w.r.t. estimation errors of distributionally robust soft actor-critic (DR-SAC) algorithm (Alg. 2) against soft actor-critic (SAC) (Haarnoja et al., 2018b). We experiment on continuous control tasks from PyBullet simulation module (Coumans & Bai, 2016–2019). In particular, we focus on Hopper and Walker2D domain that exhibit the most variance during training. We note that our scores are not directly comparable to the ones reported in (Haarnoja et al., 2018b) since the Roboschool environments behind the PyBullet module are harder than the MuJoCo Gym environments.

We build our implementation on top of the Softlearning package[1]. We use the same hyperparameters as described in (Haarnoja et al., 2018b). In addition, we set parameters $C = 1$ and $\eta = 0.5$. We plan to study the impact of hyperparameter choice in future work.

To implement per-state counter $n_t(s)$, we discretize the state space as follows. First, we discretize each dimension into ten equal-size bins, since each dimension takes values from a bounded range. Then, the state representation is given by a joint representation of its dimensions. We count the number of times the state representation appears along the policy trajectory. We leave the question of finding good state representation for future research.

First, we analyze robustness during training. Fig. 1 shows mean and standard deviation of metrics computed over 5 evaluation rollouts at each training step using stochastic policy. Each evaluation rollout spans over 1000 environment steps. We perform 5 runs of each algorithm with different random seed. The solid curves corresponds to the mean and the shaded region to the minimum and maximum returns over the 5 trials. As expected, the DR-SAC algorithm greatly reduces variance during training in terms of standard

[1]https://github.com/rail-berkeley/softlearning

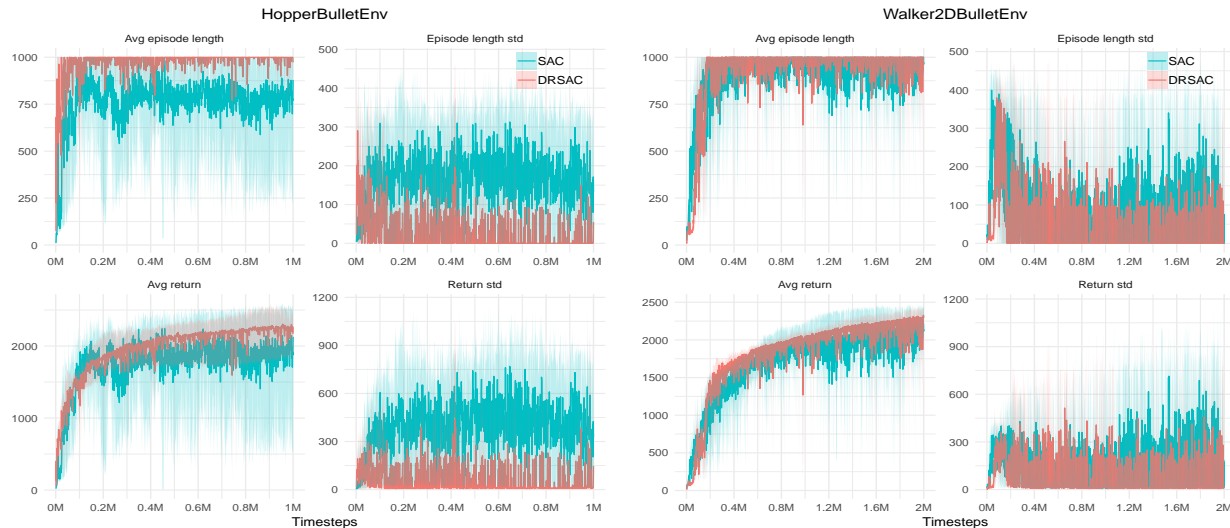

*Figure 1.* Average and standard deviation of return and episode length during training on Hopper and Walker2D domains. DR-SAC shows significant reduction of variance of return and episode length.

deviation of return and episode length. We note that the absolute value of average return is similar to the one of SAC algorithm, but the upper confidence bound shows clearly lower. This empirically confirms the safety guarantee provided by the DR-SAC.

Next, we analyze the final policy performance. Table 1 presents evaluation results computed across 100 test trajectories over stochastic policies. The policies produced using DR-SAC algorithm generate trajectories with significantly smaller variance of return and episode length. The mean performance and episode length do not show statistically significant difference. Thus, DR-SAC achieves smaller variance of performance without decreasing the average value.

Finally, the video demonstrations of learned policies are available online[2]. Qualitatively, the movements of DR-SAC policies are slower and less abrupt than the ones of SAC policies.

## 7. Conclusion

We study the risk-averse exploration in approximate RL setting. We propose the distributionally robust modified policy iteration scheme that implements safety in policy evaluation step w.r.t. estimation errors. The proposed scheme is based on a family of distributionally robust Bellman operators that provide tight lower bound guarantee on policy state-values. From a theoretical perspective, we establish the convergence of our scheme to the optimal policy. Practically, the proposed scheme results in tractable algorithms both in the

discrete and continuous settings. The proposed practical algorithm implements a mixed exploration strategy that ensures safety at short-term and encourages exploration at long-term. Our experimental results show that distributional robustness is a promising direction for improving stability of training and ensuring the safe behaviour RL algorithms. In future work, we plan to extend the experimental evaluation to more tasks and provide guidance on the choice of hyperparameters.

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

# A. Appendix

## A.1. Convergence of distributionally robust MPI

**Lemma 2.** *Let $\pi$ and $\pi'$ be two policies, and define $\Delta\pi :=$ $\pi'(\cdot|s) - \pi(\cdot|s)$, for every state $s \in \mathcal{S}$. Then*

$$|(\mathcal{T}^{\pi'}V)(s) - (\mathcal{T}^{\pi}V)(s)| \leq (R_{\max} + \gamma\|V\|_\infty)\|\Delta\pi(\cdot|s)\|_1.$$

*Proof.* We first note that $\pi \mapsto \mathcal{T}^\pi$ is Lipschitz w.r.t the TV distance on policies. By direct computation, one has

$$
\begin{aligned}
&|(\mathcal{T}^{\pi'}V)(s) - (\mathcal{T}^{\pi}V)(s)| \\
&= |r^{\pi'}(s) + \gamma P^{\pi'}(s,\cdot)^T V - r^\pi(s) - \gamma P^\pi(s,\cdot)^T V| \\
&= |(\pi'(\cdot|s) - \pi(\cdot|s))^T r(s,\cdot) + (P^{\pi'}(s|\cdot) - P^\pi(\cdot|s))^T V| \\
&\leq |(\pi'(\cdot|s) - \pi(\cdot|s))^T r(s,\cdot)| + |\Delta\pi(\cdot|s)P(\cdot|s,\cdot)V| \\
&\leq \|\Delta\pi(\cdot|s)\|_1 \|r(s,\cdot)\|_\infty + \|\Delta\pi(\cdot|s)\|_1 \|P(\cdot|s,\cdot)^T V\|_\infty \\
&\leq (R_{\max} + \gamma\|V\|_\infty)\|\Delta\pi(\cdot|s)\|_1.
\end{aligned}
$$

where in the last but one inequality, we have used the fact that $\|P(\cdot|s,\cdot)V\|_\infty := \max_{a\in\mathcal{A}} |P(\cdot|s,a)^T V| \leq \max_{a\in\mathcal{A}} \|P(\cdot|s,a)^T V\|_1 \|V\|_\infty = \|V\|_\infty$, because $\|P(\cdot|,s,a)\|_1 = 1$ for all $a \in \mathcal{A}$ since $P$ is a transition matrix. $\square$

**Theorem 1** (Distributionally robust modified policy iteration). *For an integer $m \in [1, \infty]$, consider the distributionally robust MPI scheme (18), where*

$$
\epsilon_t(s) = \begin{cases} Cn_t(s)^{-\eta}, & \text{if } n_t(s) \geq t/S, \\ 0, & \text{else,} \end{cases}
$$

*for some constants $C, \eta > 0$ and $S$ denotes the number of states.*

*Let $\tilde{\ell}_t := \tilde{V}_t - V^* \in \mathbb{R}^\mathcal{S}$ be the loss at iteration $t$. Then after $N \geq 2$ iterations, we have*

*(A) **Sub-optimality bound:***

$$\|\tilde{\ell}_N\|_\infty \leq \frac{4R_{\max}}{(1-\gamma)^2} E_N + \frac{2\gamma^N}{1-\gamma}\|\tilde{\ell}_0\|_\infty,$$

*where*

$$E_N := \sum_{t=1}^{N-1} \gamma^t \|\delta_{N-t}\|_\infty = \mathcal{O}_N\left(\frac{CS^\eta}{(1-\gamma)N^\eta}\right) \longrightarrow 0.$$

*(B) **Safety guarantee:***

$$\tilde{V}_t \leq V_t \leq V^*, \ \forall t \in \{1, \ldots, N\},$$

*where $V_t$ is the value function computed via exact evaluation step (5).*

*Proof.* *Part (A)* Let $\pi$ and $\pi'$ be two policies, and set $\Delta\pi := \pi'(\cdot|s) - \pi(\cdot|s)$, for every state $s \in \mathcal{S}$. Then, given any value function $v \in \mathbb{R}^\mathcal{S}$ and applying Lemma 2, one has

$$|(\mathcal{T}^{\pi'}V)(s) - (\mathcal{T}^{\pi}V)(s)| \leq (R_{\max} + \gamma\|V\|_\infty)\|\Delta\pi(\cdot|s)\|_1.$$

Further, using Pinsker's inequality, one has

$$|(\mathcal{T}^{\pi'}V)(s) - (\mathcal{T}^{\pi}V)(s)| \leq (R_{\max} + \gamma\|V\|_\infty)\sqrt{D_{KL}(\pi'\|\pi)}. \quad (29)$$

Now, using the inequality (29) and the definition of $\tilde{V}_t$

$$
\begin{aligned}
0 \quad &\leq ((T^{\pi_t^{\epsilon_t}})^m \tilde{V}_{t-1})(s) - \tilde{V}_t(s) \\
&= ((\mathcal{T}^{\pi_t^{\epsilon_t}})^m \tilde{V}_{t-1})(s) - ((T^{\pi_t})^m \tilde{V}_{t-1})(s) \\
&\leq (R_{\max} + \gamma\|\tilde{V}_{t-1}\|_\infty)\sqrt{D_{KL}(\pi_t^{\epsilon_t}(\cdot|s)\|\pi_t(\cdot|s))} \\
&\leq \|\tilde{V}_{t-1}\|_\infty (1-\gamma)^{-1} R_{\max}\epsilon_t(s),
\end{aligned}
$$

where, in the last inequality we have used the fact that $\|\tilde{V}_{t-1}\|_\infty \leq (1-\gamma)^{-1} R_{\max}$. On the other hand, by construction of the errors $\epsilon_t$, one has

$$\|\epsilon_t\|_\infty = \max_s \epsilon_t(s) = \max_{s|n_t(s) \geq t/S} Cn_t(s)^{-\eta} \leq CS^\eta t^{-\eta}.$$

Thus, setting $\square := (1-\gamma)^{-1} R_{\max}$, for large $N$, one can bound $E_N$ as follows:

$$
\begin{aligned}
E_N &= \sum_{t=1}^{N-1} \gamma^t \|\delta_{N-t}\|_\infty \leq \square \sum_{t=1}^{N-1} \gamma^{N-t} \|\epsilon_t\|_\infty \\
&\leq \square CS^\eta \gamma^N \sum_{t=1}^{N-1} \gamma^{-t} t^{-\eta} \sim \frac{CS^\eta}{(1-\gamma)N^\eta},
\end{aligned}
$$

where the last asymptote is via this MathOverflow post https://mathoverflow.net/q/329893. The desired result then follows from Lemma 1.

*Part (B).* Follows from definition of $\pi^{\epsilon_t}$. $\square$