# OpenReview forum: "Distributionally Robust Reinforcement Learning"
_ICML.cc/2019/Workshop/RL4RealLife — RL4RealLife 2019_

### Official Review · AnonReviewer1 · 2019-05-15
**Nice idea; derivations are confusing at times**

**Rating:** 2
**Confidence:** 3

**Review:**

The authors propose a distributionally robust policy iteration algorithm.  Value iteration is performed with respect to an adversarial policy (rather than the current policy).  The per-state adversarial policy is chosen to be a policy close to the current policy (measured by a KL-divergence between action distributions) while being worst in terms of backed up value.  The authors augment this scheme with an entropy regularizer and adapt it to continuous control domains.  Preliminary results suggest that the proposed algorithm can achieve similar good reward performance while reducing variance of performance in training.

Some comments:
-- The idea is interesting and seems potentially fruitful.
-- The text contains many grammatical errors.  I suggest the authors perform a detailed review of the writing.
-- The motivation behind Eq 14 is unclear.  Initially the text suggests that policy iteration should be robust to unknown dynamics.  But this equation provides robustness w.r.t. policy action distribution.  What is the relation?
-- Equation 11 and the notation leading up to it is very unclear.  Is this written properly?  If I substitute in the definitions, I get Tv + \eps <= Tv <= Tv + 2\eps, which does not make sense.
-- Similarly, Equation 18 is very unclear.  The equation does not correspond to the text (prev sentence) which presents it: "The adversarial Bellman operator (14) can be expressed as a regularized Bellman operator (8) w.r.t. adversarial policy (16)"
-- The algorithm bears some similarity to entropy-regularized RL algorithms with automatically-tuned regularization; see https://arxiv.org/abs/1707.01891 and https://arxiv.org/abs/1812.11103
-- As the authors note in the abstract, the experiments are very preliminary.  I encourage the authors to add results on more domains.  More detailed analysis of the algorithm behavior (e.g. on simpler tasks) may help as well.

Overall, the paper in its current form is difficult to evaluate mostly due to the lack of clarity.  If the authors take time to improve the clarity of the derivations and writing, I am confident that a future revision will be more worthy of presentation.

---

### Official Review · AnonReviewer2 · 2019-05-23
**Substantial theoretical practical contributions to risk-averse RL, highly relevant for the workshop topic.**

**Rating:** 5
**Confidence:** 2

**Review:**

The paper presents substantial contributions related to a topic of high interest in applying reinforcement learning to real-world problems. Safe exploration is studied from both a theoretical and practical point of view by providing proofs and visualizations of the learned behaviors.

The authors could describe how practically feasible is to guarantee (12) happens in practice.

The theoretical contributions are rather involved and while they are clearly constructed, the less familiar reader would benefit from a more presentation of the intuition behind the different constructs. E.g. can (13) be constructed in practice?

In my opinion, the paper should be presented and discussed in detail at the workshop, as risk-averse exploration is strongly desired and oftentimes deployment of RL algorithms is prevented due to lack of guarantees for worst-case scenarios. Further continued experiments similar to the ones provided on the Hopper task would be useful to quantify the practical applicability of the algorithm.

---

### Decision · Program_Chairs · 2019-05-28

Accept